# A Method of Predicting Wear and Damage of Pantograph Sliding Strips Based on Artificial Neural Networks

**DOI:** 10.3390/ma15010098

**Published:** 2021-12-23

**Authors:** Małgorzata Kuźnar, Augustyn Lorenc

**Affiliations:** Department of Rail Vehicles and Transport, Cracow University of Technology, 31-878 Cracow, Poland; alorenc@pk.edu.pl

**Keywords:** pantograph sliding strip, AI for prediction, artificial neural network, damage prevention, predictive maintenance

## Abstract

The impact of the pantograph of a rail vehicle on the overhead contact line depends on many factors. Among other things, the type of pantograph, i.e., the material of the sliding strip, influences the wear and possible damage to the sliding strip. The possibility of predicting pantograph failures may make it possible to reduce the number of these kinds of failures. This article presents a method for predicting the technical state of the pantograph by using artificial neural networks. The presented method enables the prediction of the wear and damage of the pantograph, with particular emphasis on carbon sliding strips. The paper compares 12 predictive models based on regression algorithms, where different training algorithms and activation functions were used. Two different types of training data were also used. Such a distinction made it possible to determine the optimal structure of the input and output data teaching the neural network, as well as the determination of the best structure and parameters of the model enabling the prediction of the technical condition of the current collector.

## 1. Introduction

Reliability and safety of railway vehicles is strongly connected with the correct power reception from the overhead contact line.

Direct contact of the rail vehicle and catenary system is possible thanks to carbon sliding strips placed on the current collector’s slider.

This can be a replaceable element attached to the base of the slider or it can be integrated with the base of the slider. Failure of this rail part may result in damages to the catenary line. The cost of this kind of incident is very high, and the result could be dangerous.

In case of replacement caused by damage while the vehicle is running, steps are taken to assess criteria, such as:material of sliding strips melting in case of arcing and damages caused by an electric arc (not proper contact force);tearing off of a piece of carbon sliding strip;cracks on the surface penetrating a sliding strip;peeling off the top layer of a sliding strip in direct contact with the catenary.

The cause of damage to the sliding strips depends on many factors related to regulations of the pantograph, such as contact force between current collector and catenary, but also the conditions of rail vehicle operation play a huge role. Because of this, the air temperature and humidity, as well as ice on the catenary lines during the winter season, should be taken into account. The additional factors related to the production, such as material defects of the sliding strips, and related with the quality of the railway infrastructure should also be covered.

On the technical review the technical condition of the pantograph is checked, but on the primary review, P1 is not measured, and the visual analyses of the technical state are carried out. On the P2 the pantograph and carbon strip is measured and noticed in the paper card.

Due to the impact of many factors on the failures of the current collector, typical mathematical modelling or linear programming cannot be used to predict the technical state of the sliding strip. To solve this problem the heuristics method and Artificial Intelligence method may be used for making the model.

In Poland, until 2011, copper sliding strips were mainly used, after which the obligation to use carbon sliding strips on current collectors of traction vehicles was introduced for carriers using the Polish traction network. This was due to the necessity to adapt to the provisions contained in the Technical Specifications for Interoperability (TSI) [1,2,3,4].

The contact strip material should be mechanically and electrically adjusted to the physicochemical properties of the contact wire material (in accordance with clause 4.2.18 of TSI [3]). This avoids, among other things, excessive abrasion of the contact wire surfaces and the contact strips themselves.

Replacing copper sliding strips with carbon was used to limit the occurrence of unfavourable phenomena such as [5]:intense abrasive wear and peeling of the contact wire and the sliding strip surface;spot erosion;the extraction and transfer of material particles or the deposition of molten metal condensate.

In the case of using copper sliding strips, the phenomenon of the accumulation of wear products in the grease contained in the slider and the grease residues remaining on the contact wire was also observed. This had a negative effect on the wear rate of the sliding strips and the contact wire.

The materials that conduct electricity and have the best lubricating properties include suitable mixtures of graphite (carbon). As a result of the strong adhesion of graphite particles to the metal surface, and as a result of the cooperation of the sliding strip made of carbon-based material, a graphite layer is applied to the contact wire surface. It reduces the friction coefficient between the sliding strip and the contact wire, increasing resistance to higher temperatures, improving the conductivity at the contact point and in a visibly way reducing the wear of both the contact strip and the contact wire. For these reasons, carbon sliding strips have been used in Poland since 2011.

Many factors contribute to the characteristics of the wear process. These are, among others, the operating conditions (temperature, humidity, wind), features of the means of transport (stiffness of wheelsets, speed of movement), features of the current collector (stiffness, dynamics, dirt, force difference when lifting and lowering, contact force, current value, condition of contact strips), overhead contact line (line tension, icing, mechanical damage). Hence, many interrelated factors need to be analysed. Such cases are called NP-hard problems, unable to be solved by typical linear optimization. To solve this type of problem, it is indispensable to use tools that can take into account many parameters related to each other.

### 1.1. Wear Pantograph Sliding Strips

The problem of interaction between the overhead contact line and the current collector is related to numerous scientific works, the selected list of which from recent years has been included in items [6,7,8,9,10,11,12,13,14,15] of the attached bibliography. In many of them, attention is drawn to the fact that to maintain an uninterrupted contact of the slider with the catenary, maintenance of appropriate pressure force of the current collector is necessary.

In the case of too much contact force, the intensity of abrasive wear increases and this may lead to mechanical damage of the current collector or its parts or the overhead contact line. In the case of too low force, favourable conditions are created for the formation of an electric arc, which causes the burning of the carbon sliding strip.

During the regular use of the current collector, the contact strip wears out naturally. If the thickness of the sliding strip is smaller than the value assumed in the measurement card, the sliding strip should be replaced with a new one. Many companies prefer to replace the strips a little earlier to protect against the situation in which such an exceeding of the limit value may occur and cause costly damage to the current collector and the overhead contact line. Such a replacement of the sliding strip without obvious signs of damage is a sanctioned practice in the system of rail vehicle operation. There are no specific clear criteria for making such a decision. Otherwise, the cause of replacement is various types of damage. The most frequent types of damage to carbon overlays includes cracks in the overlay, detachment of a fragment of the overlay, peeling off the top layer of a strip, material melting as a result of arcing and cracking of the strip.

### 1.2. State of the Art

At the turn of the 20th and 21st centuries, scientists began research on the railway pantograph and overhead catenary system. One of the first papers presenting the basic mathematical model of contact between sliding strips and overhead contact line was published by Abbot [16] in 1970. A more detailed model was presented in 2015 by Wilk et al. [17,18]. Abdullach et al. [6,7] paid attention in their research to the integration of two separated models: the catenary model and the pantograph model. Pisano and Usai [19] analysed problems in the high speed train transportation systems connected to the quality of pantograph. Tao et al. [20] and Ding et al. [21] researched tribology behaviours of a carbon strip and copper wire under electric currents. Many of the presented studies work mainly on the mathematical methods of simulation of dynamic [16,17,18,22,23,24,25] and analysis of contact force [6,7,9,19,26,27,28,29], as well as on the wear of sliding strip material [20,21,30,31,32,33,34,35,36]. The papers focus on material properties and the interaction between sliding strips and catenary. This research was hard to implement for companies because of a lack of data—with their services data being stored in the paper form and having low computing force—and lack of tools. The situation has changed with the advancement of computing technology and digitization. One of the most important pieces of research was published by Bruni et al. [37]. In this research, 10 of the simulation models were used for the study of the interaction between pantograph and catenary. One static and three dynamic simulation examples were presented for a high-speed couple (pantograph and catenary). The impact of regulation on the quality of the current collector is presented by Song Y. et al. [38]. The authors proposed preprocessing procedure to eliminate unnecessary information in measurements irregularities based on the EEMD (ensemble empirical mode decomposition); then, they analyzed the information on the position of the dropper on the contact line. The presented results show that random irregularities have a direct impact on the pantograph–overhead contact line, including the contact force statistics. Zhang et al. [39] made the review of the previous works focuses on the dynamic characteristics of the pantograph and catenary parts, the dynamic system properties, and the environmental influences on the pantograph-catenary interaction. The newest research, presented in 2021 by Song et al. [40], proves the importance of taking into account for simulation both models: pantograph–catenary and vehicle–track. The quality of the presented models was done by experimental test and the world benchmark.

In the last 20 years, the popularity of using neural networks in engineering and technical diagnostics and maintenance has rapidly grown. Scientists deal with the optimization of resources to reduce wastes and costs and increase reliability. Gajewski and Vališ [41] in their research focus on prolonging the operation time of the engine. They investigated data using the synergistic method combining the computing powers of enhanced decision trees and artificial neural networks. Moreover, Kuzhagaliyeva et al. [42] focused their work on the effectiveness of engines. For modern engines, there is a trend of downsizing and boosting it, but it is important at least keep it running efficiently or also increase it. To do this, detecting pre-ignition is very important. This is performed by using in-cylinder pressure sensors with high sensitivity and precision, but is also expensive. Authors propose the convolutional neural networks (CNNs) in object detection combined with recurrent neural networks (RNNs). The authors present developed models for pre-ignition detection based on lambda sensors (the lower cost than typical sensors with high precision). Other research [43] used an Artificial Neural Network to modify the values output signals for prompt maintenance and current repairs as well as the safe operation of the vehicle. This solution has self-diagnostic solutions. Furthermore, similar methods can be used for the analysis of the technical state of the vehicle part. Borecki et al. [44] proposed using Multi-layer Perceptron (MLP) and Kohonen Self-Organizing Maps (SOM) for analyses of the technical condition of the wheel rim. For a different technical object—batteries—Khaleghi S. et al. [45] proposed recurrent neural networks for online systems for analyses filling the battery.

The rapid growth of technology, artificial intelligence tools, sensors, Big Data analysis and cloud computing contributed to the development of the idea of predictive maintenance. Predictive maintenance is used to find a solution to predict and prevent the faults in the different machines. Salini et al. [46] developed the methodology for predicting the occurrence of critical Diagnostic Trouble codes. Their model is also based on the Convolutional Neural Network (CNN) and Fully Connected (FC) layer framework. The method enables the achievement of the average accuracy of 72% cases. Also, predictive maintenance is used for the condition of the technical equipment of vehicles. Wen Y. et al. [47] used statistical and artificial intelligence methods for machine prognostics for predictive maintenance. In this area, Bhat J. et al. [48] proposed using DR ferrography predictive maintenance of worm gearbox. Also for engines, similar research can be found—[49] used PREPIPE to predict oxygen sensor clogging conditions, to control combustion efficiency and pollutant emissions.

In the area of predictive maintenance, Cao Q. et al. [50] developed software to enhance production efficiency and reliability. They used, for the core of this software, Artificial Intelligence technologies—machine learning and data mining—to predict potential anomalies within manufacturing processes.

## 2. Materials and Methods

The data for this research was acquired from one of the biggest rail carriers in Poland. The data from the technical review are put into a measurement card. As part of the preliminary research, data was collected from measurement cards (written manually by employees of entities responsible for the operation of rail vehicles) and information was collected from field experts regarding the types of damage to the current collectors. This allowed for a certain assessment of the scale of the problem of replacing carbon contact strips in relation to the entire current collector. The following information was recorded in the measurement card:time of raising the current collector to the rated value;fall time of the current collector;correct control of the collectors from both cabins, correct movement of the collector;average static pressure;force difference when lifting and lowering;holding force measurement (folded);checking the degree of wear of the contact inserts of the slider;insulation resistance measurement.

A total of about 750 cards with over 1500 measurements of current collectors made over a period of 2 years were analyzed. The measurements concerned 62 locomotives types EP09 and EU07. These cards show that during the P2/P3 maintenance (at the P2/P3 “maintenance level” according to DSU), 8.3% of the current collectors were replaced in full (127 cases), while in 273 cases the sliding block was replaced, which accounts for as much as 17.8% of all cases. The analysis of the collected data shows that in the enterprise, the most serviced were vehicles with AKP-4E collectors (56.6%), and vehicles with 5ZL collectors constituted 29.9%. It follows that 86.5% of the serviced current collectors were four-arm collectors, and only 13.5% were single-arm DSA 150 collectors.

### 2.1. Causes of the Replacement of the Sliding Strip and Its Thickness

The information about the causes of the replacement of the sliding strip is not noted on the measurement cards, so we created algorithms to identify the reasons for the replacement. These pre-analyzed data were used for developing the prediction model.

Reliability assessment and knowledge of experts allowed us to prepare identification algorithms for technical conditions and for replacement causes. The exemplary algorithms are shown below [51]:(1)$$Wop=1\Leftrightarrow Nl_{i+1}=Nl_{i}\;\wedge (Top_{i+1}\neq \;Top_{i}\;\vee \;Nop_{i+1}\neq \;Nop_{i})$$
(2)$$Wn=1\Leftrightarrow \left(Nl_{i+1}=Nl_{i}\right)\wedge (Top_{i+1}=Top_{i})\wedge (Wop_{i}\neq 1)\wedge \;((Gn1_{i}-\;Gn1_{i+1}<0)\vee (Gn2_{i}-Gn2_{i+1}<0))$$
(3)$$N_{1}=1\Leftrightarrow Wn=1\;\wedge N_{3}=0\;\wedge (Gn1<32\vee Gn2<32)$$
(4)$$N_{2}=1\Leftrightarrow Wn=1\;\wedge \left(N_{1}+N_{3}=0\right)\wedge ((Gn1>33)\vee (Gn2>33))$$
(5)$$N_{3}=1\Leftrightarrow Nop=1\wedge \left(\left|Gn1-Gn2\right|\geq 2\right)$$
where:*Wop* replacement of the pantograph*Wn* replacement of the pantograph sliding strip*Nl* the locomotive number*Nop* the pantograph number*Top* the type of pantograph*Gn1* thickness of the first carbon sliding strip*Gn2* thickness of the second carbon sliding strip*N1* replacement of the sliding strip due to even wear of the sliders*N2* replacement of the sliding strip due to detachment of a fragment of the sliding strip, material extraction or burning of the sliding strip*N3* replacement of the sliding strip due to uneven wear of the sliders*i* the measure number

The results of the analysis of the relationship between the causes of the replacement of the sliding strip and its thickness are presented below. The tests were carried out for two current collectors, each with two sliding strips.

Figure 1 and Figure 2 show the reason for replacing the first and second sliding strip of pantograph A, considering the thickness of the strip during replacement. Wear of the sliding strip is the most common cause of strip replacement when it is less than 31 mm thick. However, the minimum value of the cover thickness is 23 mm for the AKP-4E and 5ZL current collector and 25 mm for the DSA 150 collector. The analysis clearly shows that the replacement of the cover is carried out too early. For strip thickness above 31 mm, the most common reason for replacing the sliding strip turns out to be damage. The concept of damage in this analysis includes damage caused by detachment of a fragment of the sliding strip, peeling off the top layer of a strip, material melting as a result of arcing and cracking of the strip. In this range of thicknesses, the entire current collector is also frequently replaced.

Figure 3 and Figure 4 show the same analysis of the dependence of the cause of replacement of the sliding strip on the strip thickness, but in this case, they concern the pantograph B. Both for collectors A and B, the most frequent replacement occurs when the thickness of the sliding strip is in the range of 30 to 32 mm.

Based on Figure 1, it was noticed that for the sliding strips with a thickness lower than 37 mm, unexpected damage could occur. Because of this, many sliding strips are replaced near this value of thickness. That is the reason why the number of damages grows for strips with a thickness over 39 mm. It was also noticed that for thicknesses from 38 mm the number of pantograph damages grows. For thicknesses lower than 32 mm, the sliding strips are always replaced, which is the reason why the number of damages was reduced to zero.

For sliding strip number 2 of pantograph A, the conclusion could be very similar those for sliding strip number 1. Up to the 32 mm thickness, there is a growing risk of damaging the sliding strip, which means there is also a risk of damage the pantograph. Because of this, the service procedure requires the replacement of the sliding strips with lower than 32 mm thickness.

The analysis of pantograph B (Figure 3 and Figure 4) confirm the conclusions from pantograph A. As pantograph A and B are used alternately, this depends on the driving direction of the rail vehicle. For the thickness, anything lower than 36 or 37 mm to 32 mm, the number of damages of sliding strips and also pantograph grows. After 32 mm, the sliding strips are always replaced and so the reason for damage (if unexpected damages did not occur before) is always wear.

### 2.2. Wear and Damage Prediction Method

To select a model to identify the technical condition of the current collector sliding strip, which takes into account the degree of wear of the strip and the reason for its replacement, prediction models were developed and tested with the use of artificial neural networks.

Several combinations of training datasets were also tested. Table 1 shows selected sets of input data (Input), while Table 2 presents the structure of the output data (Target/Output).

In Table 2 we present three technical conditions (able to further use, limited ability of further use, not able to further use). Typically there are just two states (able to further use, not able to further use). We propose a prediction model which predicts the three technical states. State three means that pantograph cannot be used and the cause of this could be connected with wear or with damage. The model can predict the wear but the damage is occasionally caused by accidents, so it is not predicted by the model. In these cases, the pantograph damage is in state three—not able to be further used. 

To prepare the input and output data, data from the measurement cards were organized, analyzed and then the final set of data had to be determined. As the season and the type of collector affect the frequency of replacements in preliminary studies, this information has been included in the dataset. The final structure of the database was established based on establishing the relationship between the collected data and the technical condition and reasons for replacing the sliding strips. Based on the knowledge of the destructive processes operating during the use of the current collector, preliminary research and the knowledge of experts in the field of current collectors, decision-making rules have been developed to identify the causes of replacement of the sliding strips. Statistical analysis was performed to determine the relationship between the data from monthly measurements and the reasons for the replacement. 

The sets of input and output training data have been differentiated not only due to the search for the best structure of such data but also due to the prediction methods that require a slightly different approach. It was decided to use an artificial neural network as a supervised machine learning method to build predictive models. For machine learning classification methods, there can be only one answer which is called a Response, while for regression methods, there can be multiple answers—in this case, multiple Outputs. The presented sets of input and output data are applicable with the use of regression methods, i.e., with the use of artificial neural networks used in the article. In the case of the classification method, there must be only one answer in the case of the output data, while in the case of the regression method (i.e., the artificial neural networks used in the article), both single and multiple responses can be used. As part of this article, research was carried out only on regression models based on artificial neural networks. The classification method using decision trees was described in the article [51].

### 2.3. Summary of Predictive Models

Due to the large number of studies conducted, it was decided to include only representative models. This article compares 12 predictive models based on artificial neural networks. Figure 5 shows schematically the process of creating a model during training and the process of predicting results based on the developed model. Table 3 lists 12 predictive models.

The predictive models presented in Table 3 shows the main models from conducted research to achieve the final model—number 12. The final model gave the best results of prediction. The models presented in the table are based on the previous one but in the upgraded version. Changes between models based on the experience of authors in the fields of neural networks, data processing and simulation. Conducted research enabled the achievement of the neural network structure with good prediction results.

In Table 3, for each of the models, the type of the applied machine learning method is presented, in accordance with Table 4. The given input data (Input/Predictors) were consistent with Table 1, while the output data (Output/Response) were consistent with Table 2.

The sets of input and output training data have been differentiated not only due to the search for the best structure of such data but also due to individual prediction methods that require a slightly different approach.

Table 4 summarizes the tested types of artificial neural networks. This compilation includes models with the use of regression algorithms and supervised learning. These models were compiled in tabular form with assigned appropriate names, so as not to duplicate information about the type of learning, learning category, type of neural network and properties such as activation function or training algorithm when describing predictive models.

The first type of training was supervised by the regression method with the help of artificial neural networks of the feed-forward type, where the tangensoid activation function was used for all network layers. The Levemberg–Marquardt training algorithm was used in this case.

The second of the neural networks presented in the table differs from the first in the activation function—both tangensoidal and linear activation functions were used in this case. A linear activation function was applied to the output layer in this case.

The third type of artificial neural network is also the feed-forward type with backward propagation, where, similarly to network number 2, tangensoidal and linear activation functions are used. In this case, however, Bayesian regularization was used as the training algorithm.

In the fourth of the presented types of networks, two different training algorithms were used. The Levemberg–Marquardt training algorithm was used in the first step of the training, while the Bayesian regularization was used in the second step.

The fifth tested method of artificial intelligence, like the previous ones, belongs to the regression of supervised learning. In this case, the applied neural network of the feed-forward type contained a time delay (Feed-forward distributed time delay). The tangensoid activation function was used herein all layers, while Bayesian regularization was adopted as the training algorithm.

The sixth method was also a time-delayed feed-forward neural network, but in this case, the training algorithm was based on the method of gradients coupled with the Powell–Beale algorithm.

The remaining tested models, developed on the basis of regression algorithms, did not give satisfactory results, therefore they were not included in the table. Nevertheless, it is worth noting that artificial neural networks were tested for such training algorithms as:Levenberg–Marquardt training algorithm (Levenberg–Marquardt back-propagation TRAINLM);Bayesian Regularization back-propagation TRAINBR;Riedmiller and Braun algorithm RPROP (Resilient Backpropagation -TRAINRP);Steepest Descent Algorithm (TRAINGD);Gradient Descent Algorithm with Moment—TRAINGDM;The method of gradients coupled with the Powell-Beale algorithm (TRAINCGB);The method of gradients coupled with the Polak-Ribier algorithm (TRAINCGP).

The next chapter discusses the process of selecting the appropriate neural network structure, while Figure 6 shows a diagram of the process of teaching a predictive model (artificial neural network), based on which the technical condition of the current collector was predicted.

## 3. Results

### 3.1. Examined Prediction Models

To develop the final predictive model, in the next steps, artificial neural networks were modified in terms of training algorithms, activation functions in individual layers, the number of layers and the structure of input and output data. The purpose of these modifications was to develop a model based on artificial neural networks that would enable the best results of the prediction of technical states of the current collector.

In the beginning, the training data sets were divided into two groups—Data set A and Data set B, where set A contained at least three instances of maintenance in the cycle (i.e., from replacement to the next replacement of the sliding pad), while data set B contained all technical reviews.

Model 1 was a feed-forward artificial neural network with backpropagation. According to Table 3, it contained five hidden layers, with 14 neurons in each layer. In this case, the Tangensoidal activation function and the Levenberg–Marqardt training algorithm were used. During the training and simulation of dataset A, the correct classification of state 2 for this model was only 4.3%. During the training and simulation of the data set with all technical inspections (data set B), the correctness of the classification increased to 20.5%, however, the correctness of the classification of state 3 decreased significantly, from 83.0% to 57.8%.

Model 1 did not meet the expectations related to the correct determination of individual technical conditions. Therefore, several studies have been carried out, which changed, inter alia, learning algorithm, activation functions or the number of hidden layers. None of the models made it possible to exceed the 30% correctness of state 2 predictions. Therefore, it was decided to check a different structure of the input data. Another prediction model presented was developed based on the second set of input data in accordance with Table 1.

In the case of model 2, despite a significant improvement in the prediction of state 2, the tested model did not meet the expectations related to the correct determination of individual technical states. During the training and simulation of the data set, where the cycle included at least three reviews (Data A), the correct classification of state 2 was 31.9%, but the correctness of the classification of state 3 was 0%. In this case, it was decided not to conduct research on the second data set—Data B—because previous studies showed a reduction in the correctness of the state 3 classification for Data B against Data A.

In the next step, the same network was retrained to see how it would affect the results. The maximum number of steps for the learning algorithm was changed from six to five.

Despite the re-training of Model 2, as well as the change in the maximum number of steps of the learning algorithm, the tested Model 3 did not meet the expectations related to the correct determination of individual technical conditions. For the training and simulation process, a data set where there were at least three reviews per cycle (Data set A), the correct classification of state 3 was greater than that of Model 2, however, the correctness of the classification of state 2 decreased to 10.6%. For the analogous ANN structure, research was carried out by changing individual training parameters, as well as changing the network learning algorithms and activation functions, but no better results were obtained.

Therefore, it was decided to change the structure of the neural network under study.

In the next step, the neural network available in the “Neural Fitting app” was examined. It is a Feed-forward Artificial Neural Network with backpropagation implemented in the Matlab Machine Learning module.

Model 4 used the Levenberg–Marqardt training algorithm. This model had only one hidden layer with 12 neurons, where the Tangensoidal activation function was used, while the output layer used a linear activation function. The change in the ANN structure, as well as the activation function in the output layer, increased the correctness of the classification of state 2 and state 3. However, a decrease in the correctness of the classification of state 1 was noted. Therefore, in the next step, the number of neurons in the hidden layer was changed from 12 to 6.

In the case of Model 5, where the number of neurons in the hidden layer was reduced compared to Model 4, an increase in incorrect classification for states 2 and 3 was noticed. State 1 was correctly classified in 92.7%. Nevertheless, the tested model did not meet the expectations related to the correct determination of individual technical conditions. In the next step, it was decided to change the structure of the input data for the third set of data presented in Table 1, and change the number of neurons in the hidden layer to be equal to the number of input data.

The change in the structure of both the Artificial Neural Network and the input data resulted in a significant improvement in learning outcomes. In the case of model 6, as much as 76.7% of all technical conditions were classified correctly. The second condition was correctly classified in 38.8%. Due to a significant improvement in the results, in the next model it was decided to keep the network structure, but to change the training algorithm to Bayesian regularization.

The results of the correctness of the classification of technical conditions in the case of Model 7 turned out to be the same as in the case of Model 6, in which the Levenberg–Marqardt training algorithm was used. The model learner time in the case of model 7 was, however, much longer than in the case of model 6.

Despite the similar results of both models, it was decided to re-test the same network structure by combining both learning methods. In Model 8, due to the speed of training, in the first step, it was decided to use the Levenberg–Marqardt algorithm. Then Bayesian regularization was used because it is a modified Levenberg–Marqardt algorithm that has better generalization properties. Due to the properties of both training algorithms, different results were expected than in the case of training the models 6 and 7.

Testing model 8 had a positive effect. The correct classification of state 2, in this case, was 42.6%. In the next step, it was therefore decided to examine a neural network with a similar structure, but a different type—Feed-forward time-delay.

Distributed delay networks are similar to feed-forward with backpropagation except that each input and layer weight has a delay line associated with it. As a result, the network has a finite dynamic response to the input of the time series. This network is also similar to a time-delayed neural network (timedelaynet) which only has delays on the input weight.

Model 9, being a feed-forward neural network with distributed time-delay, turned out to give better results both during training and simulation, compared to the analogous network with backpropagation. This shows that the sequence and continuity of measurements during technical inspections is important for the prediction of technical conditions. To check whether the network was overtrained, the tested number of epochs was reduced from 1000 in Model 9 to 100 in Model 10.

Reducing the number of epochs resulted in a slight difference in the parameters for evaluating the correctness of the network training process in relation to the network with a greater number of epochs. Nevertheless, reducing the number of epochs significantly shortened the training time, and also increased the accuracy of the classification of state 3, which was 100%.

In the next step, an analogous network with a time delay was tested, but with a different structure of the input data. In subsequent models, the output data of the 1 × 1 structure was used, where the value in the range [1,2,3] determined the technical condition (the second set of outputs from Table 2 was used). Among the networks tested, the best one was the one in which the method of gradients coupled with the Powell–Beale algorithm was implemented. Model 11 was developed on this basis.

This model turned out to perform better than model 10 in both training and simulation. This shows that not only the sequence and continuity of measurements during technical inspections is important for the prediction of technical conditions, but also the structure of the output data.

To check the significance of the time delay in the case of such a defined output data structure, in the next step, the network implemented in the Matlab “Neural Fitting app” module was examined.

Thus, in Model 12, a feed-forward network with backpropagation was used, and a linear activation function was used for the output layer. As a result, the correctness of the classification of state 2 was as high as 85.1%. Among all tested predictive models, Model 12 turned out to be the best in terms of the correct classification of technical conditions.

Further ANN research, so the modifications to, inter alia, structure and parameters of the network, did not allow us to obtain better results both during training and simulation. The results obtained from model 12 were considered satisfactory due to the high correctness of the classification of state 2, allowing for appropriate maintenance activities to minimize damage to the sliding strips and current collectors.

### 3.2. Results of the Prediction Model

Table 5 provides information on the evaluation of individual predictive models. To evaluate the learning process of artificial neural networks, the Mean Square Error (MSE) and the R correlation coefficient (standard Persona correlation coefficient for the set value and the value obtained at the network output) were used.

To evaluate the simulation process, a method was developed to determine the correctness of the classification of technical conditions. For this purpose, the prediction results of each of the technical conditions were compared and the final technical condition obtained as a result of the prediction was considered to be most likely to occur. Outputs from regression methods returned in this case by neural network occur as a decimal value (in a continues form). The expected result, so the technical condition, have to be presented in a district form (whole value). Because we tested various types of neural networks as an example of the regression method, it was necessary to convert the data from continuous to discrete form. Results obtained from the simulations carried out using neural networks were divided into appropriate three classes. Thanks to this procedure, the simulation results, which were in a continuous form, obtained a discrete form and could be compared to a real technical condition which was divided into three classes. Equations (1)–(3) show the ranges of values that enable assigning a given continuous value to one of the three classes. This example is presented for model 12, but for other models the transformation between continuous and discrete form was similar—it was based on the determination of appropriate intervals for belonging to a given class.

In the next step, the simulation results transformed into discrete forms were compared with the real ones. As a result, the percentage correctness of the classification of each technical condition was obtained. Table 5 presents the results of the assessment of the correctness of the classification of all technical conditions and the correctness of the classification of the second technical condition $S_{2}$ as the key condition for the possibility of preventing damage to the sliding cover of the current collector.

Among the tested predictive models, the highest correctness of the classification of second technical condition $S_{2}$ was achieved for Model 12. It was as much as 85.1%. Moreover, the correctness of classification of all technical conditions turned out to be the highest for this model (82.5%). In this case, however, relatively low correctness of the classification of state 1 was obtained, which amounted to 62.4%.

In terms of learning results, Model 9 turned out to be the best. In this case, the mean square error MSE was only 0.014731, and the correlation coefficient was as high as 0.9222. The results obtained for the training process turned out to be the best for Model 9. The correctness of the classification of technical conditions, in this case, was, however, lower than for Model 12 and amounted to 78.6% for all technical conditions and 61.7% for the second technical condition $S_{2}$.

In the case of simulations, with the use of models based on regression algorithms, a histogram of errors was presented, which defines the differences between the values obtained from prediction and the actual values of the technical condition, and a graph of the correctness of the classification of technical conditions.

The results of the prediction using model 12 give the result as one value in the range from 0 to 3.5. For this model, thresholds for the technical conditions were defined experimentally:(6)$$S_{1}\Leftrightarrow \left[0\leq y_{12}<1.25\right]$$
(7)$$S_{2}\Leftrightarrow \left[1.3\leq y_{12}\leq 2.5\right]$$
(8)$$S_{3}\Leftrightarrow \left[2.5<y_{12}\leq 3.5\right]$$
where:

$S_{1}$—First technical condition—able to further use$S_{2}$—Second technical condition—the limited ability of further use, it will be necessary to replace the sliding strip for the next inspection$S_{3}$—Third technical condition—not able to further use—it is necessary to replace the sliding strip/pantograph$y_{12}$—the value obtained during the prediction thanks to the use of themodel 12.

Due to the determination of the presented ranges of limit values determining the expected technical condition, it was possible to determine the correctness of the classification of technical conditions. For this purpose, the percentage share of $\;y_{12}$ values correctly classified to technical conditions were calculated. Therefore, the correctness of the classification means the number of cases classified to the same technical condition as in reality.

Figure 7 shows model 12 consisting of two layers, where the tangensoidal activation function was used in the first layer, and the linear activation function was used in the more expensive layer. The Levemberg–Marquardt training algorithm was used in this model.

Figure 8, Figure 9, Figure 10 and Figure 11 concern the training process of the artificial neural network constituting model 12. The regression function fit for all processes during training an ANN (training, validation and test) was R = 0.90966 (Figure 8). This function for the entire network training process took the form of:(9)Output=0.83 · Target+0.28

The regression function for training was R = 0.91632, for Validation R = 0.84508, while for the test process it was R = 0.94019. The mean square error in this model was 0.15974 and was reached in the 12th epoch (Figure 9).

Figure 10 presents selected ANN error parameters for subsequent learning epochs. As you can see, the gradient for the learning epoch 18 is 0.1293. The Mu-factor for this epoch was 0.0001. Figure 11 is an error histogram for the training process showing the difference between the targets and the output values from the network. The chart shows errors for the neural network training, testing and validation process. Most cases oscillate in the range from −0.1778 to 0.07845.

Figure 12 and Figure 13 relate directly to the simulation process. Figure 12 shows a histogram of errors, where most cases fluctuate in the range from −0.3068 to 0.01282. Figure 13 shows the correctness of the classification of all three technical conditions. The correctness of the classification is presented in percentage terms, and it is based on formulas (1)–(3).

The correctness of the classification of technical conditions was as follows:

for technical condition $S_{1}$: 62.4%;for technical condition $S_{2}$: 85.1%;for technical condition $S_{3}$: 100%.

To reduce the damages of the current collectors, the most important factor is to identify technical state “2”. This state means that the analysed object need to be soon replaced by a new one. Therefore, it means that for identifying this state with a high probability—85.1% it will be possible to detect low-quality parts before damage occurs or the minimum value of the thickness of the sliding strip will be exceeded. Practically speaking, these results are very useful for the railway carrier because, as of now, the internal company rules to decide on the replacement of sliding strips are based just on the thickness of the sliding strips and the experience of service staff. In many cases, this replacement was too fast and caused the additional cost and waste of the components that could be used for longer. What is more, in some cases, replacement is too late and causes the damage of the whole current collector, so the costs of a new one are major. The service companies do not have a similar prediction system to help decide about the optimal time of sliding strips replacement.

## 4. Discussion

By analysing the results of both training and simulation, it can be concluded that Model 12 best meets the expectations related to the correct classification of the second technical condition $S_{2}$.

Model 12 uses the same input and output data structure as in Model 11. However, in the case of Model 12, a feed-forward network with backward propagation was used, and a linear activation function was used for the output layer. As a result, the correctness of the classification of state 2 was as high as 85.1%.

Among the tested predictive models, Model 12 turned out to be the best in terms of the correct classification of technical conditions. Further ANN studies did not allow to obtain better results both during training and simulation.

The main drawback of the presented model is a classification of the 37.6% of cases from technical condition S1 to S2. This situation does not have a negative effect on the unpredicted wear or damage of sliding strips, but causes the earlier replacement of the sliding strips—the predicted state is S2, so it suggests the high possibility of wear before the next service. Because of this, the model suggests a replacement. It does not cause additional costs in current maintenance services, because now the sliding strips are generally replaced earlier than they could be. So, savings are not able to be made. It is obvious that future research and development of better models to enable to prediction of the correct state 1 with a higher probability will increase savings and reduce costs.

## 5. Conclusions

Based on the conducted research, it can be concluded that artificial neural networks can be successfully used to predict the wear and damage of carbon sliding strips. The best prediction results were obtained for the input data set, which included parameters such as review number, information about a new measuring cycle, number of days since the exchange, a quarter of the year, pantograph type, information about pantograph position (front/rear pantograph), difference in thickness of the strip N1 and N2 between inspections, information about the earlier technical condition, and if there was pantograph replacement information about the reason. There was no information in this dataset regarding average temperature in the month, average wind speed for the month and total rainfall for the month. It can be concluded that these data were too general for the prediction of the technical condition of the pantograph, but the introduction of more accurate weather data would probably increase the correctness of the classification of technical conditions. Such a conclusion can be made due to the significant improvement of the prediction results after adding to the input data the quarter of the month in which the technical inspection was performed.

The form of the output data is also important for predicting the technical condition of the pantograph. It should be noted that presenting the output data as values (z_3_(t)), and not as a vector T, significantly increased the correctness of the classification of the second technical condition $S_{2}$.

The best learning algorithm turned out to be the Levemberg–Marquardt algorithm, where both tangensoidal and linear activation functions were used. Further research on the influence of the learning algorithm and the activation functions did not show any improvement in the correctness of the classification of technical conditions.

In subsequent studies, it should be assumed that the correctness of the classification may be influenced by such factors as more accurate weather conditions, data on the route. Based on the currently available data, an increase in the correctness of the classification of technical conditions may occur as a result of combining several predictive models and adjusting the appropriate threshold values for individual classes of technical conditions.

## Figures and Tables

**Figure 1 materials-15-00098-f001:**
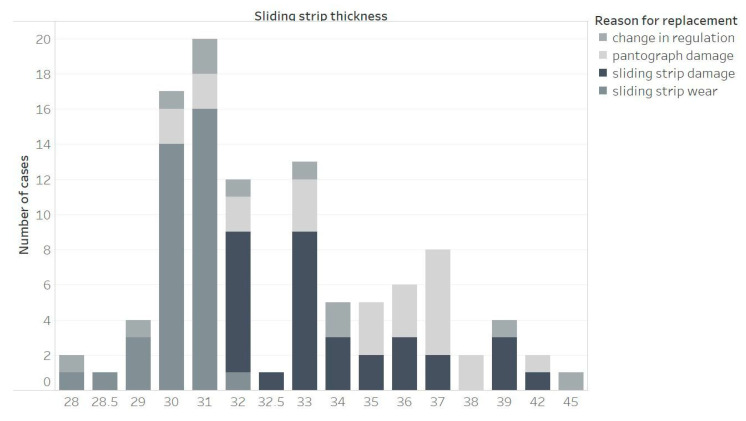
Reason for replacing sliding strip number 1 of pantograph A, depending on the thickness of the sliding strip.

**Figure 2 materials-15-00098-f002:**
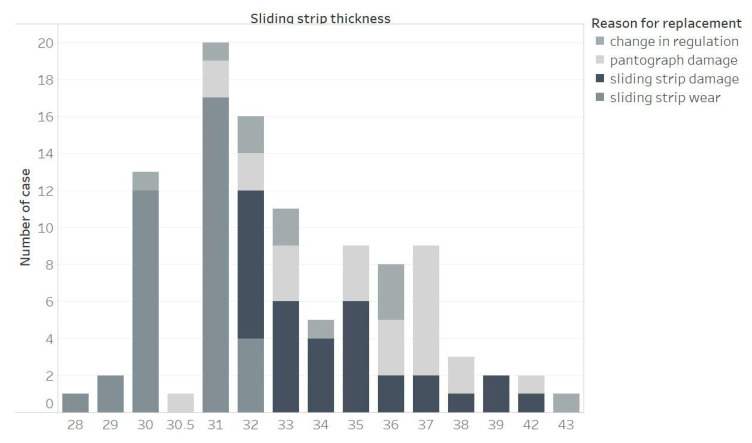
Reason for replacing sliding strip number 2 of pantograph A, depending on the thickness of the sliding strip.

**Figure 3 materials-15-00098-f003:**
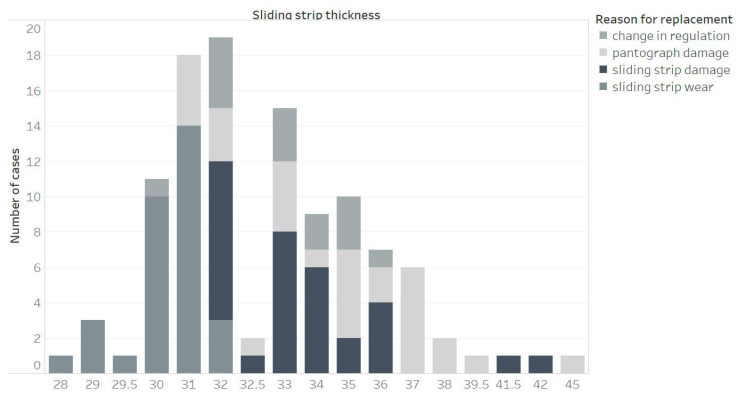
Reason for replacing sliding strip number 1 of pantograph B, depending on the thickness of the sliding strip.

**Figure 4 materials-15-00098-f004:**
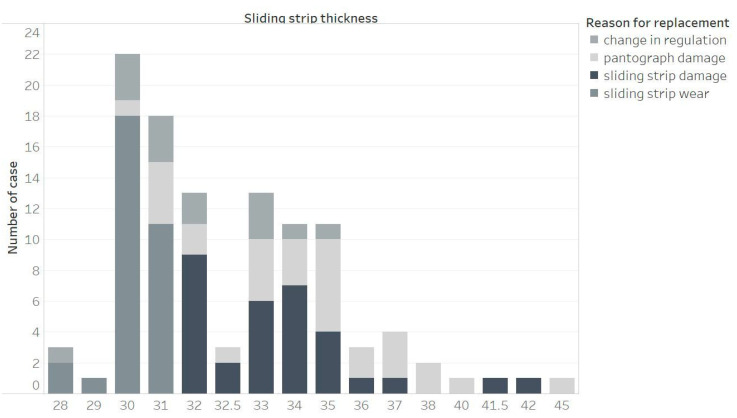
Reason for replacing sliding strip number 2 of pantograph B, depending on the thickness of the sliding strip.

**Figure 5 materials-15-00098-f005:**
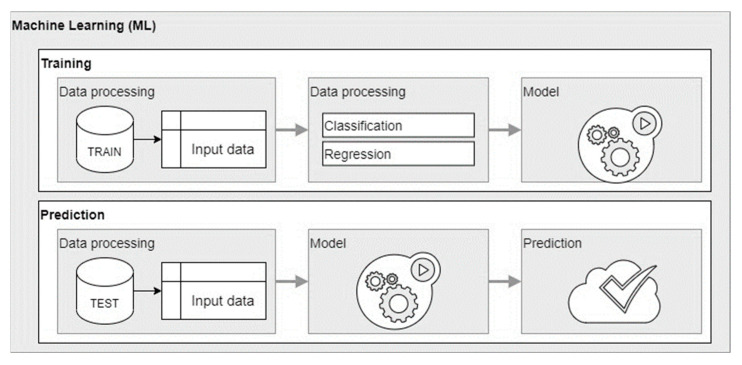
Diagram of the model creation and prediction process.

**Figure 6 materials-15-00098-f006:**
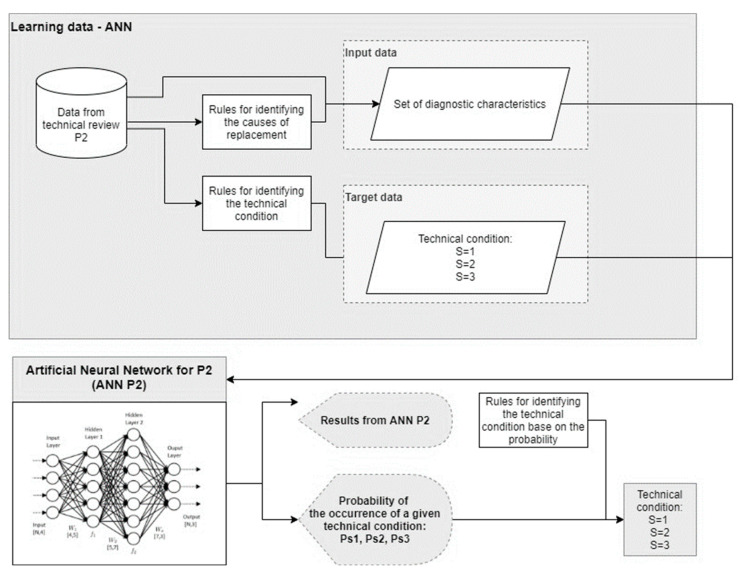
Diagram of the training process of predictive models.

**Figure 7 materials-15-00098-f007:**
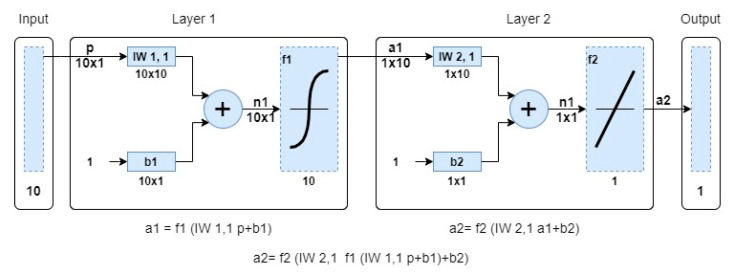
Artificial Neural Network—Model 12.

**Figure 8 materials-15-00098-f008:**
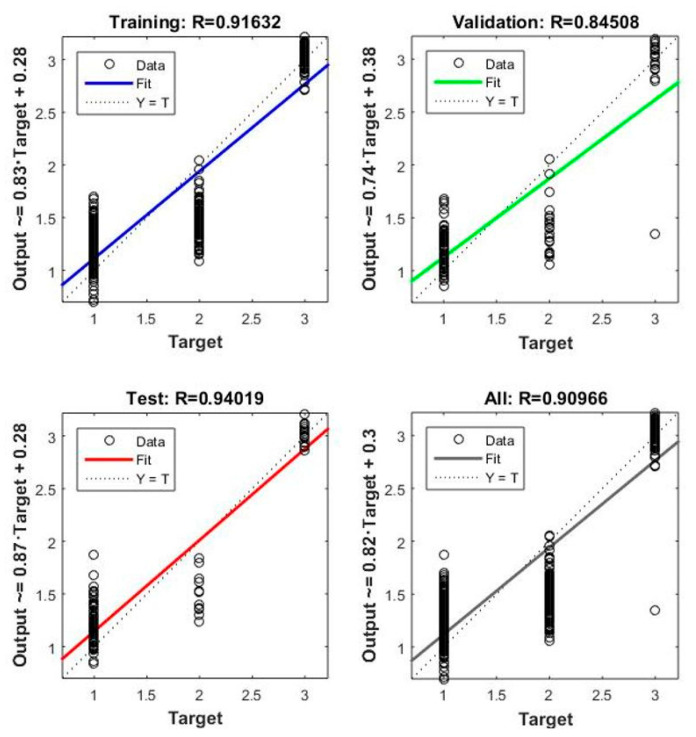
Regression function for the learning process—Model 12.

**Figure 9 materials-15-00098-f009:**
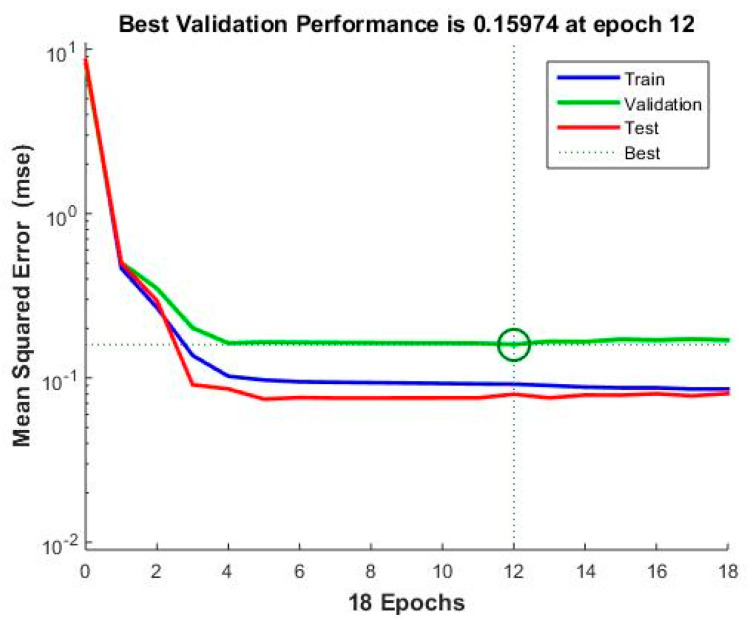
The smallest mean square error for the learning process—Model 12.

**Figure 10 materials-15-00098-f010:**
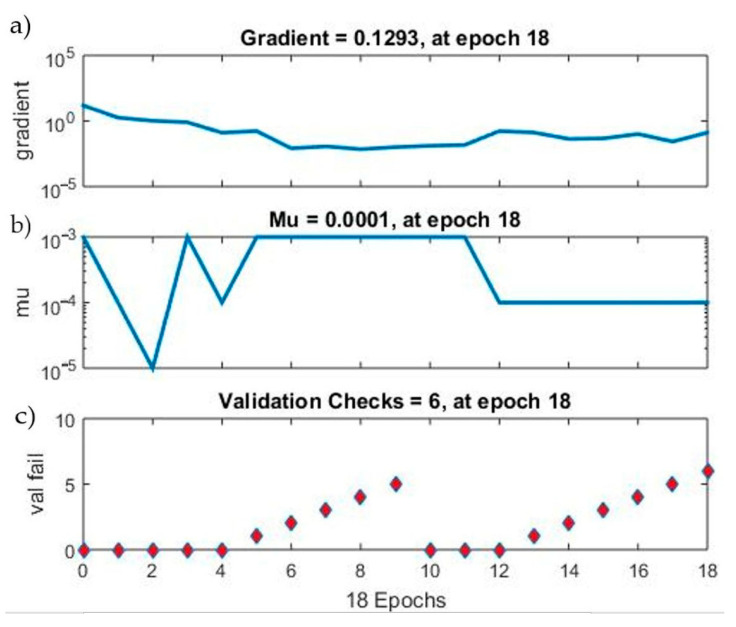
A course of network learning errors for a given epoch: (**a**) minimal error of the network training gradient; (**b**) Mu coefficient; (**c**) validation check—Model 12.

**Figure 11 materials-15-00098-f011:**
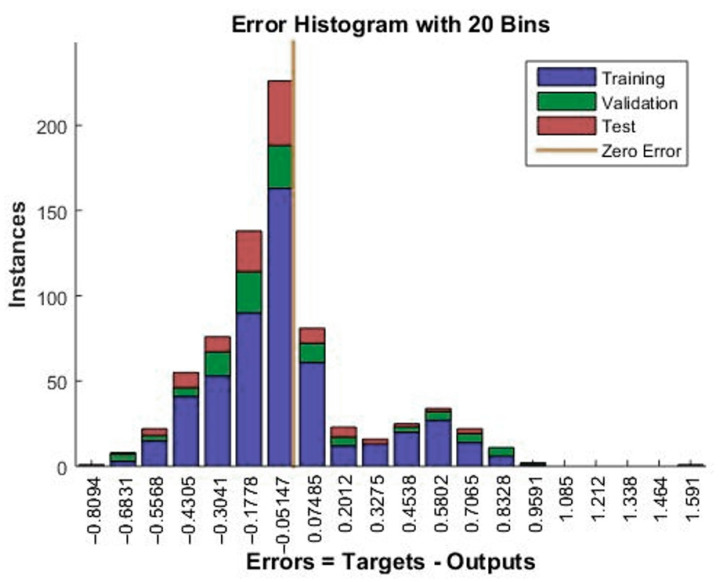
Diagram Error histogram for the learning process—Model 12.

**Figure 12 materials-15-00098-f012:**
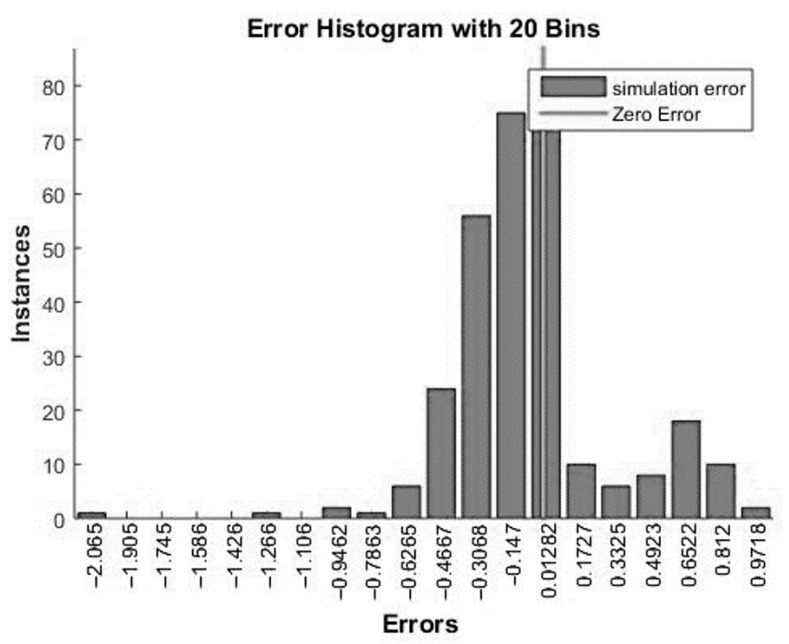
Error histogram for the simulation process—Model 12.

**Figure 13 materials-15-00098-f013:**
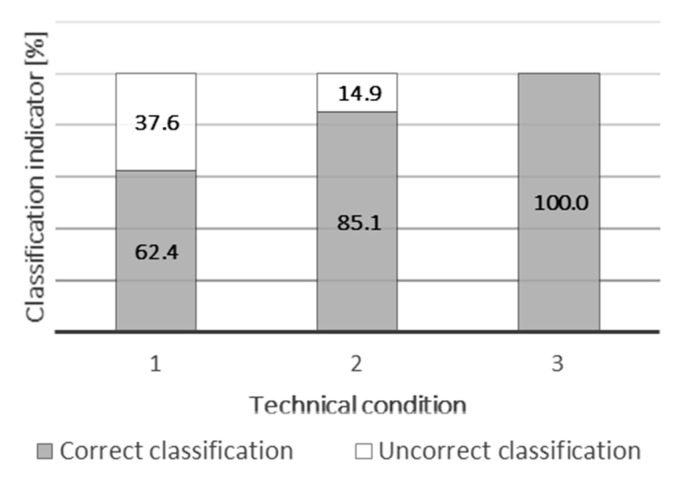
Correctness of classification of technical conditions—Model 12.

**Table 1 materials-15-00098-t001:** Input data for a prediction model.

Type of Input Data		Input Data Structure Number		
		1	2	3
The number of input data s_in_		14	12	10
1	Review number	X	X	X
2	New measuring cycle	X	X	X
3	The number of days since the exchange	X	X	X
4	Quarter of the year	X	X	X
5	Average temperature in the month [°C]	X	X	
6	Average wind speed for the month [km/h]	X	X	
7	Total rainfall for the month [mm]	X	X	
8	Pantograph type	X	X	X
9	Front/rear pantograph	X	X	X
10	The difference in thickness of the strip N1 between inspections	X	X	X
11	The difference in thickness of the strip N2 between inspections	X	X	X
12	Sliding strip thickness N1	X		
13	Sliding strip thickness N2	X		
14	Reason for replacement during the previous measurement	X	X	
15	Earlier technical condition			X
16	Reason for replacement			X

**Table 2 materials-15-00098-t002:** Output data.

The Type of Output		The Output Data Structure Number	
		1	2
The number of outputs s_out_		3 vector T	1 value z_3_(t)
1	First technical condition—able to further use (S_1_ = 0 lub S_1_ = 1)	X	
2	Second technical condition—the limited ability of further use, it will be necessary to replace the sliding strip for the next inspection (S_2_ = 0 or S_2_ = 1)	X	
3	Third technical condition—not able to further use—it is necessary to replace the sliding strip/pantograph (S_3_ = 0 lub S_3_ = 1)	X	
4	Technical condition z_3_(t) specified as value: z_3_(t) = 1 First technical condition z_3_(t) = 2 Second technical condition z_3_(t) = 3 Third technical condition		X

**Table 3 materials-15-00098-t003:** List of selected predictive models.

Model No.	Type of Learning Method	Input/Predictors Regards to Table 1	Output/Response Regards to Table 2	Model Parameters
1	(1) ANN F-T-Lm	1	1	Number of hidden layers: 5 (14-14-14-14-14-3)
2	(1) ANN F-T-Lm	2	1	Number of hidden layers: 5 (12-12-12-12-12-3)
3	(1) ANN F-T-Lm	2	1	Number of hidden layers: 5 (12-12-12-12-12-3)
4	(2) ANN F-TP-Lm	2	1	Number of hidden layers: 1 (12-3)
5	(3) ANN F-TP-Lm	2	1	Number of hidden layers: 1 (6-3)
6	(2) ANN F-TP-Lm	3	1	Number of hidden layers: 1 (10-3)
7	(3) ANN F-TP-Br	3	1	Number of hidden layers: 1 (10-3)
8	(4) ANN F-TP-Lm/Br	3	1	Number of hidden layers: 1 (10-3)
9	(5) ANN Ft-T-Br	3	1	Number of hidden layers: 1 (10-3)
10	(5) ANN Ft-T-Br	3	1	Number of hidden layers: 1 (10-3)
11	(6) ANN Ft-T-C	3	2	Number of hidden layers: 1 (10-3)
12	(2) ANN F-TP-Lm	3	2	Number of hidden layers: 1 (10-3)

**Table 4 materials-15-00098-t004:** List of selected types of artificial intelligence methods.

No.	Name	Type of Model/Neural Network	Properties	
1	ANN F-T-Lm	Feed forward artificial neural network with backpropagation	Activation function: TANSIG	Learning algorithm: TRAINLM
2	ANN F-TP-Lm	Feed forward artificial neural network with backpropagation	Activation function: TANSIG/PURELIN	Learning algorithm: TRAINLM
3	ANN F-TP-Br	Feed forward artificial neural network with backpropagation	Activation function: TANSIG/PURELIN	Learning algorithm: TRAINBR
4	ANN F-TP-Lm/Br	Feed forward artificial neural network with backpropagation	Activation function: TANSIG/PURELIN	Learning algorithm: TRAINLM/TRAINBR
5	ANN Ft-T-Br	Feed forward artificial neural network with backpropagation distributed time-delay	Activation function: TANSIG	Learning algorithm: TRAINBR
6	ANN Ft-T-C	Feed forward artificial neural network with backpropagation distributed time-delay	Activation function: TANSIG	Learning algorithm: TRAINCGB

**Table 5 materials-15-00098-t005:** Summary of the results of selected predictive models.

Model No.	Method Type (acc. to Table 4)	Input (acc. to Table 1)	Output (acc. to Table 2)	Training		Simulation	
				MSE	R	The Correctness of the Classification of All Technical Conditions	The Correctness of Classification of the Second Condition S_2_
1	(1) ANN F-T-Lm	1	1	0.12497 (A) 0.12485 (B)	0.75943 0.64552	59.8 54.7	4.3 20.5
2	(1) ANN F-T-Lm	2	1	0.13384	0.67901	41.2	31.9
3	(1) ANN F-T-Lm	2	1	0.11970	0.71918	48.1	10.6
4	(2) ANN F-TP-Lm	2	1	0.11913	0.72838	53.2	17.0
5	(2) ANN F-TP-Lm	2	1	0.11669	0.71501	49.3	12.8
6	(2) ANN F-TP-Lm	3	1	0.069108	0.84538	76.5	38.3
7	(3) ANN F-TP-Br	3	1	0.043195	0.87944	76.2	38.3
8	(4) ANN F-TP-Lm/Br	3	1	0.038862	0.88206	78.1	42.6
9	(5) ANN Ft-T-Br	3	1	0.014731	0.9222	78.6	61.7
10	(5) ANN Ft-T-Br	3	1	0.020364	0.92088	82.0	61.7
11	(6) ANN Ft-T-C	3	2	0.064105	0.8938	81.5	80.9
12	(2) ANN F-TP-Lm	3	2	0.15974	0.90966	82.5	85.1

The designations presented in Table 5 are: MSE—Mean Square Error; R—Persona correlation coefficient.

## Data Availability

Not applicable.

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
