# Peer review of "A Method of Predicting Wear and Damage of Pantograph Sliding Strips Based on Artificial Neural Networks"

_materials, 2021, doi:10.3390/ma15010098_

Round 1

Reviewer 1 Report

Generally, the prediction of pantograph sliding strip wear is quite important to ensure a safe and reliable operation of a pantograph-catenary system. The topic of this paper is quite important. Some comments are suggested as follows for better improving this paper.

Firstly, in the literature review, when describing others' works about pantograph-catenary simulation and analysis of contact force, more classic works published in prestigious international papers deserve to be mentioned. Some of them are suggested as follows. Among them, [1] is the most important work in the last decade as it directly compares the results of ten leading software. [2] is the first attempt to study the spatial dynamics of whole railway system. [3] firstly reports the stochastic irregularities, which are directly relevant to the present paper. [4] is a review of the research about pantograph-catenary interaction. The reviewer thinks these works are much more worthwhile to be cited.

[1] S. Bruni, J. Ambrosio, A. Carnicero, Y.H. Cho, L. Finner, M. Ikeda, S.Y. Kwon, J.P. Massat, S. Stichel, M. Tur, W. Zhang, The results of the pantograph-catenary interaction benchmark, Veh. Syst. Dyn. 53 (2015) 412–435. https://doi.org/10.1080/00423114.2014.953183.

[2] Y. Song, Z. Wang, Z. Liu, R. Wang, A spatial coupling model to study dynamic performance of pantograph-catenary with vehicle-track excitation, Mech. Syst. Signal Process. 151 (2021) 107336. https://doi.org/10.1016/j.ymssp.2020.107336.

[3] W. Zhang, D. Zou, M. Tan, N. Zhou, R. Li, G. Mei, Review of pantograph and catenary interaction, Front. Mech. Eng. 13 (2018) 311–322. https://doi.org/10.1007/s11465-018-0494-x.

Secondly, this paper included a number of types of real-life data. These data are quite important to determine the service performance of a pantograph-catenary system. But it is not quite understandable how these data were acquired. Please give a brief introduction to clarify.

Thirdly, based on the reviewer's understanding, another important factor that may affect the wear process of the pantograph strip is the uplift force (or moment) of the pantograph, which directly affects the mean contact force. Normally a high mean contact force may lead to severer wear in the contact surface. The uplift force has some uncertainties and may not always be regulated to its standard value. But no information regarding this has been described. Please give some necessary comments on this issue.

Fourthly, the reviewer is a bit confused about how do you classify 'pantograph damage' in these three conditions in table 2. As the pantograph damage should not be counted as the pantograph wear, as it may be caused by some occasional accidents. Please comment.

The reviewer does not have any comments on the results and the main methodology of this work. Generally, the attempt in this paper is quite promising. This paper can be considered to be published after the above issues are properly tackled.

Author Response

Dear Reviewer, thank you very much for your valuable comments and suggestions. We have revised the paper accordingly to them. Detailed responses to the comments and suggestions are as follows.

Comment 1. Firstly, in the literature review, when describing others' works about pantograph-catenary simulation and analysis of contact force, more classic works published in prestigious international papers deserve to be mentioned. Some of them are suggested as follows. Among them, [1] is the most important work in the last decade as it directly compares the results of ten leading software. [2] is the first attempt to study the spatial dynamics of whole railway system. [3] firstly reports the stochastic irregularities, which are directly relevant to the present paper. [4] is a review of the research about pantograph-catenary interaction. The reviewer thinks these works are much more worthwhile to be cited.

[1] S. Bruni, J. Ambrosio, A. Carnicero, Y.H. Cho, L. Finner, M. Ikeda, S.Y. Kwon, J.P. Massat, S. Stichel, M. Tur, W. Zhang, The results of the pantograph-catenary interaction benchmark, Veh. Syst. Dyn. 53 (2015) 412–435. https://doi.org/10.1080/00423114.2014.953183.

[2] Y. Song, Z. Wang, Z. Liu, R. Wang, A spatial coupling model to study dynamic performance of pantograph-catenary with vehicle-track excitation, Mech. Syst. Signal Process. 151 (2021) 107336. https://doi.org/10.1016/j.ymssp.2020.107336.

[3] Z. Liu, A. Ronnquist, P. Navik, Z. Liu, Contact wire irregularity stochastics and effect on high-speed railway pantograph-catenary interactions, IEEE Trans. Instrum. Meas. 69 (2020) 8196–8206. https://doi.org/10.1109/TIM.2020.2987457.

[4] W. Zhang, D. Zou, M. Tan, N. Zhou, R. Li, G. Mei, Review of pantograph and catenary interaction, Front. Mech. Eng. 13 (2018) 311–322. https://doi.org/10.1007/s11465-018-0494-x.

Answer 1. Thank you for paying attention to this paper. We add the suggested papers to the state of the art presented in the introduction.

Comment 2. Secondly, this paper included a number of types of real-life data. These data are quite important to determine the service performance of a pantograph-catenary system. But it is not quite understandable how these data were acquired. Please give a brief introduction to clarify.

Answer 2. Thank you for any inaccuracy found. In section 2.1. we add the information about the source of the data and the structure and numbers of the data set.

Comment 3. Thirdly, based on the reviewer's understanding, another important factor that may affect the wear process of the pantograph strip is the uplift force (or moment) of the pantograph, which directly affects the mean contact force. Normally a high mean contact force may lead to severer wear in the contact surface. The uplift force has some uncertainties and may not always be regulated to its standard value. But no information regarding this has been described. Please give some necessary comments on this issue.

Answer 3. Thank you for your comment. Contact force is one of the very important factors in the wear of the pantograph strip. We add proper information about this in the paragraph before chapter 2.1.

Comment 4. Fourthly, the reviewer is a bit confused about how do you classify 'pantograph damage' in these three conditions in table 2. As the pantograph damage should not be counted as the pantograph wear, as it may be caused by some occasional accidents. Please comment.

Answer 4. In table 2 we present three technical conditions (able to further use, the limited ability of further use, not able to further use). Typically there are just two states (able to further use,  not able to further use). We propose a prediction model which predicts the three technical states. State three means that pantograph can not be used and the cause of this could be connected with wear or with damage. The model can predict the wear but the damage is occasionally (by accidents), so it is not predicted by the model. In sum: the pantograph damage is in state three – not able to be further used, but it is not a state but cause of this state. We add appropriate comments after table 2.

Dear Reviewer, thank you very much for your time, professionalism and advice.

Reviewer 2 Report

This very well-researched paper contains a significant amount of information, some of which is of historical and technical interest, but the relevance of some ideas makes the paper very difficult to follow.

Line 31-83 make it difficult to understand the focus of the paper. Many good an interesting ideas are presented, but the link to the present paper topic is difficult to discern,

Line 83-98 seem like this is where the introduction to the paper begins.

Lines 100-497 seem like this should be the introduction, and are less informative to the section "Materials and Methods." It seemed as if this was a second introduction to another paper. Excellent content, just in the wrong place.

There is not introduction to set the context of  Lines 150-162, and Lines 163-167 discuss analysis results in the Methods section. The idea of "test" analysis suggests this is a materials science experimental report, not a Machine Learning  (ML) paper.  Figures 1-4 contain good information, but, again, the context and significance is not clear.

Line 180 begins the discussion that seems most relevant to the thesis of the paper.  Tables 1 and 2 provide what seems to be very important results, but the context is not clear and it is hard to understand the link between the choice of a supervised method (Line 190) and the data presented.

What seems to be impled in Section 2.2 is a discussion of the various ML methods, but simple fragments of a discussion are given and the reader has to infer what must be the series of techniques being considered and the pros/cons.

Line 201 suggests a large number of studies conducted, but the methods section does not adequately outline the various methods clearly. Having an overview would more greatly inform the value of Table 3 and the techniques used to achieve those results. In that respect, the description of the methods in Lines 210-222 is something that would be more informative, earlier (I think).

The introduction to Results , Lines 224-266 are confusing. It seems like this is where the core of the paper exists, and it comes far too late and in the wrong place.

Section 3.2 does get to key results, but again, the odd part of the paper is that regression techniques are discounted at the outset of the paper, but then results for regression are presented and discussed.

Overall, it is very difficult to tease out the significance of the work performed due to the fashion in which a variety of (very good ) information is organized.  

For this to be of value to the readers, the paper need significant modification so the organization of thought and the impact of the Neural Network ML analyses are clear.

Author Response

Dear Reviewer, thank you very much for your valuable comments and suggestions. We have revised the paper accordingly to them. Detailed responses to the comments and suggestions are as follows.

Comment 1. Line 31-83 make it difficult to understand the focus of the paper. Many good an interesting ideas are presented, but the link to the present paper topic is difficult to discern,

Answer 1. The introduction has been reworded. Subchapter 1.1 “Wear of pantograph sliding strips”  and 1.2. “Wear of pantograph sliding strips” has been separated, which systematize the relationship with the topic of work.

Comment 2. Line 83-98 seem like this is where the introduction to the paper begins.

Answer 2 Thank you for any inaccuracy found. We moved the text from lines 83-98 to the first part of the introduction.

Comment 3. Lines 100-497 seem like this should be the introduction, and are less informative to the section "Materials and Methods." It seemed as if this was a second introduction to another paper. Excellent content, just in the wrong place.

Answer 3. We change the structure of the paper. We suppose that you mean lines 100-197. After your comments, we look at the current form and decided that the text from lines 100-147 should be presented in the introduction. The lines 150-179 present the analysed data but are used in the next part of the research – developing the prediction model. Because of this we add an appropriate introduction and move it to the new subchapter 2.1. “Causes of the replacement of the sliding strip and its thickness”. The lines 196-197 present the input data for the model so it should be in the chapter methods.

Comment 4. There is not introduction to set the context of  Lines 150-162, and Lines 163-167 discuss analysis results in the Methods section. The idea of "test" analysis suggests this is a materials science experimental report, not a Machine Learning  (ML) paper.  Figures 1-4 contain good information, but, again, the context and significance is not clear.

Answer 4. The lines 150-179 present the analysed data but are used in the next part of the research – developing the prediction model. Because of this we add an appropriate introduction and move it to the new subchapter 2.1. “Causes of the replacement of the sliding strip and its thickness”.  This research is the first stage of analysing data from the measurements card – we need to know the reason for the replacement because it was not noticed on the card. We think in this form the context and significance are better presented. We also add the mathematical formula which we use to analyse the reason for the replacement.

Comment 5. Line 180 begins the discussion that seems most relevant to the thesis of the paper.  Tables 1 and 2 provide what seems to be very important results, but the context is not clear and it is hard to understand the link between the choice of a supervised method (Line 190) and the data presented.

Answer 5. Before table 1 we add information about data presented in the table. The different prediction models were examined, for example, the decision tree, the Complex Tree, Medium Tree and Simple Tree methods of machine learning. And the results prove that the machine learning method will give the best results. We add the comments before line 190.

Information has been added regarding the data set from which the input and output data set was separated. The paragraph has been redrafted to make the relationship between the chosen method and the data clearer.

Comment 6. What seems to be impled in Section 2.2 is a discussion of the various ML methods, but simple fragments of a discussion are given and the reader has to infer what must be the series of techniques being considered and the pros/cons.

Answer 6. The subsection "Summary of predictive models" has been supplemented with information on the methods used in the formula. The next chapter describes the method of selecting the next tested methods.

Comment 7. Line 201 suggests a large number of studies conducted, but the methods section does not adequately outline the various methods clearly. Having an overview would more greatly inform the value of Table 3 and the techniques used to achieve those results. In that respect, the description of the methods in Lines 210-222 is something that would be more informative, earlier (I think).

Answer 7. The text has been redrafted. We hope that the applied changes outline the various methods clearly.

Comment 8. The introduction to Results , Lines 224-266 are confusing. It seems like this is where the core of the paper exists, and it comes far too late and in the wrong place.

Answer 8. As in the previous note, the text has been redrafted to ensure the clarity of a paper. The beginning of this chapter has been moved to the previous chapter describing the methods used in the work. In the "Results" chapter, a description of the modification of the structure of the applied neural network has been added to obtain better prediction results. In this way, the method of selecting the neural networks tested in the article was presented.

Comment 9. Section 3.2 does get to key results, but again, the odd part of the paper is that regression techniques are discounted at the outset of the paper, but then results for regression are presented and discussed.

Answer 9.  The article focuses on the regression method. Various types of neural networks were used as an example of the regression method. Then, the results obtained from the simulations carried out thanks to such neural networks were divided into appropriate three classes. Thanks to this procedure, the simulation results, which were in a continuous form, obtained a discrete form. Equations 1 - 3 show the ranges of values that enable assigning a given continuous value to one of the three classes. This example is presented for the model that gives the best prediction results, i.e. for model 12.

This explanation was also added in the "Results" chapter

Comment 10. Overall, it is very difficult to tease out the significance of the work performed due to the fashion in which a variety of (very good ) information is organized.  

Answer 10. Overall, it is very difficult to tease out the significance of the work performed due to the fashion in which a variety of (very good ) information is organized.  

Dear Reviewer, thank you very much for your time, professionalism and advice.

Reviewer 3 Report

This paper presents the possibility of using artificial neural networks to predict wear and damage of the pantograph, with particular emphasis on carbon sliding strips. Two different types of training data were also used. The paper is technically sound. But the authors need to consider the following points:
1. In Page 8, Table 3 lists 12 selected predictive models. The paper needs to give the reason these models were chosen. 
2. In Page 16, the correctness of the classification for technical condition ?1 was only 62.4%. It's obvious that this correctness is not good, so how do these results demonstrate the effectiveness of this model? 
3. What are the drawbacks of the study? 

Author Response

Dear Reviewer, thank you very much for your valuable comments and suggestions. We have revised the paper accordingly to them. Detailed responses to the comments and suggestions are as follows.

Comment 1. In Page 8, Table 3 lists 12 selected predictive models. The paper needs to give the reason these models were chosen. 

Answer 1. We add before table 3 an explanation of how these predictive models were selected. We add the paragraph: The presented in table 3 predictive models show the main models from conducted research to achieve the final model – no. 12. The final model gave the best results of prediction. Each model presented in the table is based on the previous one but in the upgraded version. Changes between models are based on the experience of authors in the fields of the neural network, data processing and simulation. Conducted research enable to achieve the neural network structure with good prediction results.

Comment 2. In Page 16, the correctness of the classification for technical condition ?1 was only 62.4%. It's obvious that this correctness is not good, so how do these results demonstrate the effectiveness of this model? 

Answer 2. We add the paragraph with an explanation of these results after figure 13. The most important in terms of reducing damage to the current collectors is state 2, which means that soon it will be necessary to replace the sliding strip or the entire collector. Therefore, it means that in 85.1% of cases it will be possible to detect the need for replacement before damage occurs or the limit value of the sliding strip thickness will be exceeded. From a practical point of view, that results is very useful for the railway carrier because now the internal company rules to decide on the replacement of sliding strips based just on the thickness of sliding strips and the experience of service staff. In many cases, this replacement was too fast and causes the additional cost and waste of the components that could be used more time. What is more in some cases replacement is too late and cause the damage of the whole current collector, so the costs of a new one are major. The services company do not have a similar prediction system to help decide about the optimal time of sliding strips replacement.

Comment 3. What are the drawbacks of the study? 

Answer 3. We add the information about drawbacks to the discussion chapter. The main drawback of the presented model is a classification of the 37.6% of cases from technical condition S1 to S2. This situation does not have a negative effect on the unpredicted wear or damage of sliding strips, but cause the earlier replacement of the sliding strips – the predicted state is S2 so it suggests the high possibility of wear before the next service. Because of this, the model suggests a replacement. It does not cause the additional cost in the current maintenance services, because now the sliding strips are generally replaced earlier than they could be. So, the cause is not to make the savings. It is obvious that future research and development of better models enable to the prediction of the correct state 1 with a higher probability, and then it reduces the costs again, and the saving will be higher.

Dear Reviewer, thank you very much for your time, professionalism and advice.

Reviewer 4 Report

The paper compares 12 predictive models based on regression algorithms, where different training algorithms and activation functions were used. Two different types of training data were also used. Such a distinction made it possible to determine the optimal structure of the input and output data teaching the neural network, as well as the determination of the best structure and parameters of the model enabling the prediction of the technical condition of the current collector. This manuscript should be rejected in the current form due to lack of publication standard and novelty/originality. Please find comments as below:

  • Critical analysis should be added to the introduction to clarify the research gap/problem statement.
  • Originality and novelty are missing which requires serious attention from the authors to explain in details the new contribution to the subject in this area. 
  • "Due to the multifactor impacts to the failures of the current collector, it is impossible to use typical mathematical modelling and linear programming." What is the reason for this declaration since there are few articles in this regard? Authors are trying to show significance of their study with wrong declaration? please clarify in this regard.http://www.iaeng.org/IJAM/issues_v37/issue_2/IJAM_37_2_10.pdf
  • Detailed information should be included for figures 1-4 with respect to comparative analysis. 
  • Figure 5: Please elaborate the diagram scientifically. It is very simple in the current form. 
  • What is originality of this study other than comparative study? It looks very simple assignment to me. 
  • Results are not supported with valid literature. 
  • Comprehensive proof reading is essential throughout the manuscript.

Author Response

Dear Reviewer, thank you very much for your valuable comments and suggestions. We have revised the paper accordingly to them. Detailed responses to the comments and suggestions are as follows.

The paper compares 12 predictive models based on regression algorithms, where different training algorithms and activation functions were used. Two different types of training data were also used. Such a distinction made it possible to determine the optimal structure of the input and output data teaching the neural network, as well as the determination of the best structure and parameters of the model enabling the prediction of the technical condition of the current collector. This manuscript should be rejected in the current form due to lack of publication standard and novelty/originality. Please find comments as below:

Comment 1. Critical analysis should be added to the introduction to clarify the research gap/problem statement.

Answer 1. The introduction has been redrafted and an appropriate comment has been added to indicate a research gap.

Comment 2. Originality and novelty are missing which requires serious attention from the authors to explain in details the new contribution to the subject in this area. 

Answer 2. An appropriate comment has been added to indicate the originality and novelty of the paper. Also, reply to the 3rd comment states on this topic.

Comment 3. "Due to the multifactor impacts to the failures of the current collector, it is impossible to use typical mathematical modelling and linear programming." What is the reason for this declaration since there are few articles in this regard? Authors are trying to show significance of their study with wrong declaration? please clarify in this regard.http://www.iaeng.org/IJAM/issues_v37/issue_2/IJAM_37_2_10.pdf

Answer 3. In the article given by the reviewer, mathematical modelling was used to describe the interaction between the pantograph and the overhead contact line. For this type of description, mathematical modelling is a very common practice. Our article, however, is not about interaction modelling, but about the prediction of technical conditions and damage to the sliding pad. There are many articles on modelling interactions, as shown in the literature review, while publications related to predicting the technical condition and determining when the current collector will be damaged, unfortunately not.

In the case of only distinguishing technical states, analytical or logical models could be used, while taking into account the dependence of the technical state of the current collector on many factors (NP-hard problem) and the desire to predict such events, typical mathematical modelling does not give results.

Comment 4. Detailed information should be included for figures 1-4 with respect to comparative analysis.          

Answer 4. We add the detailed information to figures 1-4. The information includes this one above. Base on figure 1, it could be noticed that for the thickness of sliding strips lower than 37 mm, unexpected damage could occur. Because of this many sliding strips are replaced near this value of thickness. That is the reason why the number of damages is growing for thickness over 39 mm. It could be also noticed that for the thickness from 38 mm the number of pantograph damages grows. For lower thicknesses than 32 mm, the sliding strips are always replaced, that is the reason why the number of damages is reduced to zero.

For sliding strip no. 2 of pantograph A, the conclusion could be very similar to this one about sliding strip #1. To the 32 mm thickness, there is a growing risk of damaging the sliding strip and it is also a risk to damage the pantograph. Because of this, the service procedure requires the replacement of the sliding strips with lower than 32 mm thickness.

The analysis of pantograph B (figure 3 and 4) confirm the conclusions from pantograph A. Pantograph A and B are used alternately, it depends on the driving direction of the rail vehicle. For the thickness, lower than 36/37 mm to 32 mm the number of damages of sliding strips and also pantograph grows. After 32 mm the sliding strips are always replacement and so the reason for it (if the unexpected damages did not occur before) is always worn.

Comment 5. Figure 5: Please elaborate on the diagram scientifically. It is very simple in its current form. 

Answer 5. Figure 5 shows the modelling and forecasting process schematically. Such information was also included in the text.

It is a graphical representation of the process for easier visualization. The description of the procedure in a scientific form, i.e. by giving the next steps, will not make it possible and will only be a stripped-down version of what is presented in the article. The detailed diagram of the training ANN is presented in figure 6. So if figure 5 will be transformed into a more scientifically diagram it will look the same as figure 6.

Comment 6. What is originality of this study other than comparative study? It looks very simple assignment to me. 

Answer 6. The originality of the work lies in the fact that this type of research is rare in the field of research on current collectors. The comparison of the conducted research is only part of the article, which aims to show the possibilities offered by artificial neural networks, and what differences in the results can be obtained by changing some of the parameters of the neural network. The ability to predict an unwanted event and present a specific way to do so in the form of the model presented in the article makes it original. The use of the model developed in the course of these tests will make it possible to predict as much as 85.1% of cases when the sliding strips need to be replaced with a new one. Currently, this is not happening at all. The presented approach will enable the reduction of costly damage to the current collector and the traction network, as well as the reduction of train delays. It means that approach presented in the article is original, and the developed model may lead to the production of new or improved products, technological processes or organizational systems related to the diagnosis of current collectors, so this solution should be considered innovative.

Comment 7. Results are not supported with valid literature. 

Answer 7.  The results are part of the original work, and as already mentioned, this type of research, so predicting the technical condition of the current collector is a multifactorial issue and work in this area is innovative and innovative.

There is no available literature presenting the results of the prediction of the technical condition of the pantograph, so they cannot be compared with anything, but only with the actual state. In this respect, only preliminary research published by the author of this article was conducted.

Comment 8. Comprehensive proof reading is essential throughout the manuscript.

Answer 8.  The text of the article has been redrafted.

Dear Reviewer, thank you very much for your time, professionalism and advice.

Round 2

Reviewer 1 Report

All my puzzles have been properly answered by the authors. I do not have further comments regarding this paper in this round.

Reviewer 2 Report

Significant revision of the paper with dramatic improvements and updates has created a good scholarly work ready for publication.

Reviewer 4 Report

Authors have put major efforts in order to address the provided comments properly.